# Temperature Effects on Nonlinear Ultrasonic Guided Waves

**DOI:** 10.3390/ma16093548

**Published:** 2023-05-05

**Authors:** Xiaochuan Niu, Liqiang Zhu, Wenlin Yang, Zujun Yu, Haikuo Shen

**Affiliations:** 1School of Mechanical, Electronic and Control Engineering, Beijing Jiaotong University, Beijing 100044, China; xchniu@bjtu.edu.cn (X.N.); 21121281@bjtu.edu.cn (W.Y.); zjyu@bjtu.edu.cn (Z.Y.); shenhk@bjtu.edu.cn (H.S.); 2Key Laboratory of Vehicle Advanced Manufacturing, Measuring and Control Technology, Beijing Jiaotong University, Ministry of Education, Beijing 100044, China; 3Frontiers Science Center for Smart High-Speed Railway System, Beijing 100044, China

**Keywords:** nonlinear ultrasonic guided waves, temperature, cumulative second harmonics, response law, semi-analytical finite element

## Abstract

Nonlinear ultrasonic guided waves have attracted increasing attention in the field of structural health monitoring due to their high sensitivity and long detection distance. In practical applications, the temperature of the tested structure will inevitably change, so it is essential to evaluate the effects of temperature on nonlinear ultrasonic guided waves. In this paper, an analytical approach is proposed to obtain the response law of nonlinear guided waves to temperature based on the semi-analytical finite element (SAFE) method. The plate structure is investigated as a demonstration example, and the corresponding simulation analysis and experimental verification are carried out. The results show that the variation trends of different cumulative second harmonic modes with temperature are distinct, and their amplitudes monotonically increase or decrease with the continuously rising temperature. Therefore, in the applications with nonlinear ultrasonic guided waves, it is necessary to predict the changing trend of selected cumulative second harmonics under the action of temperature and compensate the result for the influence of temperature. The methods and conclusions presented in this paper are also applicable to other types of structures and have general practicality.

## 1. Introduction

In recent decades, nonlinear ultrasonic techniques have developed rapidly and have demonstrated great application prospects in the nondestructive testing (NDT) field by some researchers [1,2,3,4,5]. The nonlinear ultrasonic phenomenon indicates that, if there is a change in material properties or damage in a waveguide, in addition to the original fundamental frequency signal, ultrasonic waves will also be accompanied by the production of higher-order harmonics. High-order harmonics are much more sensitive to small changes in the structural state than conventional linear ultrasonic waves. The use of nonlinear ultrasonic characteristics can effectively measure material properties and identify damage levels.

Compared with nonlinear ultrasonic bulk waves, nonlinear ultrasonic guided waves have obvious benefits in nondestructive assessment and structural health monitoring because it combines the high sensitivity with the advantages of traditional ultrasonic guided waves [6,7,8]. Due to the above reason, nonlinear ultrasonic guided waves have received great attention in recent years, and relevant research has been carried out and made considerable strides. Liu [9] presented a non-elliptical probability imaging method based on nonlinear ultrasonic guided waves, which can be used to accurately detect the delamination damage in an anisotropic composite plate and display the damage location. Zhao [10] adopted the third harmonics of nonlinear Lamb waves to attain early fatigue damage detection in aluminum alloys. Niu [11,12] proposed a method to detect the neutral temperature in continuous welded rails by applying nonlinear ultrasonic guided waves. Hoda [13] investigated the application of nonlinear ultrasonic guided waves as a nondestructive evaluation method in detecting local corrosion of steel plates. Lee [14] studied the fatigue crack detection of steel joints subjected to tensile fatigue loading based on nonlinear ultrasonic guided waves. Yu [15] developed a new type of transducer based on nonlinear ultrasonic guided waves for detecting the damage in welded joints.

In existing studies, the influence of temperature on ultrasonic nonlinearity is not always considered in the setting of detection conditions. Since the amplitudes of high-order harmonics are very small relative to the fundamental frequency waves, the effects of temperature may cover up the change in material characteristics, which may cause a great deviation in experimental conclusions, especially in some application scenarios, such as the thermal stress detection in continuous welded rail. At present, there are few studies on the response law of ultrasonic nonlinearity with temperature.

Nucera [16] studied ultrasonic nonlinearity in constrained steel blocks under thermal stress and concluded that temperature has no effect on nonlinear ultrasonic bulk waves. Later, through theoretical derivation and experimental verification, Niu [17] demonstrated that temperature has a significant impact on nonlinear ultrasonic bulk waves. Zhao [18] studied the effects of temperature on ultrasonic nonlinear parameters in carbonated concrete and reached the same conclusion. Chillara [19] detected the change of the relative ultrasonic nonlinear parameter in a heated steel plate and found that the relative ultrasonic nonlinear parameter gradually increased with the increase of temperature. However, the research objects of articles [16,17,18] are nonlinear ultrasonic bulk waves, and article [19] lacks detailed theoretical derivation. At present, there is still a lack of exhaustive analysis about the influence of temperature on nonlinear ultrasonic guided waves.

Moreover, unlike nonlinear ultrasonic bulk waves, the main application difficulty of nonlinear ultrasonic guided waves is the excitation of desired cumulative second harmonics. The cumulative second harmonics represent the second harmonics whose amplitudes rise as the propagation distance grows. The available articles [20,21] have defined the generation conditions of cumulative second harmonics, that is, phase velocity matching and non-zero power flow. In addition to the above conditions, group velocity matching was also proposed by Bermes [22] as a necessary factor for exciting cumulative second harmonics. Deng [23] later proved that group velocity matching is not an essential condition. However, because of the multi-modal characteristics of nonlinear ultrasonic guided waves, the mode combinations that meet the preceding requirements are not single, which leads to multiple cumulative second harmonic modes in the actual experiment process. If the influence of temperature on different cumulative second harmonic modes is not identical, and several cumulative second harmonic modes are excited and overlap in the received signal, the calculation of the relative nonlinear coefficient will be jointly affected by the amplitude changes of different second harmonic modes. The above problem was also not considered in article [19]. Therefore, it is necessary to proceed with an intensive study on the variation patterns of different second harmonic modes under temperature effects.

This paper presents the systematic and comprehensive analysis of the response law of different second harmonic modes in plates with the change of temperature. Firstly, the amplitude parameter is described and introduced to characterize the amplitude of the cumulative second harmonic mode, and all possible mode combinations that may generate cumulative second harmonics are obtained. Then the response law of different second harmonic modal amplitudes under temperature is studied. Finally, relevant simulation and physical verification experiments are designed, and the simulation and experimental results are consistent with the theoretical result.

## 2. Nonlinear Ultrasonic Waves Equation and Amplitude Parameter

When ultrasonic waves propagate in the waveguide medium, it is supposed that the displacement of microscopic particles u in the medium is [24]:(1)$$u=u^{\left(1\right)}+u^{\left(2\right)},$$
where $u^{\left(1\right)}$ and $u^{\left(2\right)}$ represent particle displacement induced by the linear and nonlinear material characteristics of the waveguide, and $\left|u^{\left(1\right)}\right|>>\left|u^{\left(2\right)}\right|$. After theoretical derivation and simplification, the wave equation and boundary condition of nonlinear ultrasonic waves can be expressed as follows [25]:(2)$$\left(\lambda +\mu \right)u_{j,ji}{}^{\left(2\right)}+\mu u_{i,jj}{}^{\left(2\right)}+F^{\left(1\right)}=\rho {\ddot{u}}^{\left(2\right)},$$
(3)$$S^{L}{}^{\left(2\right)}n_{⊥}=-S^{NL}{}^{\left(1\right)}n_{⊥}\;\mathrm{on}\;S,$$
where λ,μ are the Lame’s elastic coefficients, ρ is the material density, F indicates the body force tensor, S indicates the first Piola-Kirchhoff stress tensor, $S^{L}$ and $S^{NL}$ indicate the linear and nonlinear parts of S respectively, $F^{\left(1\right)}=F\left(u^{\left(1\right)}\right)$, $S^{L}{}^{\left(2\right)}=S^{L}\left(u^{\left(2\right)}\right)$, $S^{NL}{}^{\left(1\right)}=S^{NL}\left(u^{\left(1\right)}\right)$, $\ddot{u}=\frac{\partial^{2}u}{\partial t^{2}}$, $u_{i,jj}=\frac{\partial u_{i}}{\partial x_{j}\partial x_{k}}$, i,j,k=x,y,z, $n_{⊥}$ is the unit vector of the coordinate axis, and S represents the surface of the waveguide.

Auld applied the mode expansion method to solve Equations (2) and (3) [26] and obtained the expression of the second harmonic displacement $u^{\left(2\right)}$. The formula of $u^{\left(2\right)}$ is as follows:(4)$$u^{\left(2\right)}=\sum_{n=1}^{N}A_{n}(x)u_{n}e^{-i2\omega t}+c.c.,$$
where $u_{n}$ is the mode shape of the nth second harmonic mode, N represents the number of second harmonic modes, c.c. represents the complex conjugates, x is the propagation distance, ω is the angular frequency, t is the propagation time, and $A_{n}(x)$ is the corresponding second harmonic amplitude equation.

The calculation formula of $A_{n}(x)$ is:(5)$$A_{n}(x)=\left\{\begin{array}{c} \frac{f_{n}{}^{surf}+f_{n}{}^{vol}}{2P_{nn}(k_{n}{}^{*}-2k)}\sin\left(\frac{1}{2}\left(k_{n}{}^{*}-2k\right)x\right)e^{i(k+\frac{1}{2}k_{n}{}^{*})x},k_{n}{}^{*}\neq 2k_{1}\\ \frac{f_{n}{}^{surf}+f_{n}{}^{vol}}{4P_{nn}}xe^{2ikx},k_{n}{}^{*}=2k_{1} \end{array}\right.,$$
where $P_{nn}$ is the mean energy flux density in the propagation direction, $f_{n}^{surf}$ and $f_{n}^{vol}$ represent the complex energy caused by surface stress and volume stress respectively, k is the wave number, $k_{n}{}^{*}$ is the complex conjugate of wavenumber about the nth second harmonic mode, and $k_{1}$ is the wave number of the fundamental frequency mode.

The formulas of $P_{nn}$, $f_{n}^{surf}$, and $f_{n}^{vol}$ are as follows:(6)$$P_{nn}=-\frac{1}{2}\int_{\Omega }\left(v_{n}{{}^{T}}^{*}\cdot T_{\left(n\right)}{}^{L}\right)\cdot n_{x}d\Omega ,$$
(7)$$f_{n}^{surf}=\int_{S}\left(v_{n}{{}^{T}}^{*}\cdot S_{\left(1\right)}{}^{NL}\right)\cdot \vec{n}dS,$$
(8)$$f_{n}^{vol}=\int_{\Omega }v_{n}{{}^{T}}^{*}\cdot F_{\left(1\right)}d\Omega ,$$
where $T_{\left(n\right)}{}^{L}=T^{L}(u_{n})$, $S_{\left(1\right)}{}^{NL}=S^{NL}(u_{n}{}^{\left(1\right)})$, $F_{\left(1\right)}=F(u_{n}{}^{\left(1\right)})$, $v_{n}$ is the velocity vector of the nth second harmonic mode, $u_{n}{}^{\left(1\right)}$ is the mode shape of the fundamental frequency mode, $n_{x}$ is the unit vector in the x direction, $\vec{n}$ is the unit vector perpendicular to the surface, and Ω is the cross-sectional. In the light of Equation (5), the cumulative behavior of the generated second harmonic mode occurs only when the following requirements are met [20,21]:(9)$$\left\{\begin{array}{c} Phase\;velocity\;matching:\;c_{pn}=c_{p1}\\ Non-zero\;power\;flow:\;f_{n}{}^{surf}+f_{n}{}^{vol}\neq 0 \end{array}\right.,$$
where $c_{pn}$ represents the phase velocity of the nth second harmonic mode, and $c_{p1}$ represents the phase velocity of the fundamental frequency mode. Phase velocity matching condition requires that the phase velocities of the fundamental frequency mode and the second harmonic mode be equal. A non-zero power flow condition means that the power flow transmitted from the fundamental frequency mode to the second harmonic mode is not zero. Since it is still controversial whether the group velocity matching condition needs to be satisfied to produce cumulative second harmonics, the group velocity matching condition is not studied in detail in this paper.

The second harmonic amplitude equation $A_{n}(x)$ decides the amplitude of the second harmonic mode signal, and the larger the amplitude equation, the higher the amplitude of the corresponding second harmonic mode [26]. For a certain mode combination of the fundamental frequency mode and the second harmonic mode that can generate cumulative second harmonics, when the propagation distance is fixed, the amplitude parameter of the cumulative second harmonic mode can be defined as:(10)$${\bar{A}}_{n}=\left|\frac{f_{n}{}^{surf}+f_{n}{}^{vol}}{4P_{nn}}\right|.$$
The amplitude parameter ${\bar{A}}_{n}$ can be employed to characterize the corresponding amplitude of the second harmonic mode. Under the same propagation distance, the magnitude of ${\bar{A}}_{n}$ depends on the values of $P_{nn}$, $f_{n}^{surf}$ and $f_{n}^{vol}$. In order to get the values of $P_{nn}$, $f_{n}^{surf}$ and $f_{n}^{vol}$, $S_{\left(1\right)}{}^{NL}$, $F_{\left(1\right)}$ and $T_{\left(n\right)}{}^{L}$ need to be calculated. After theoretical derivation, the calculation formula of $T_{\left(n\right)}{}^{L}$, $S_{\left(1\right)}{}^{NL}$ and $F_{\left(1\right)}$ are documented in Appendix A. From Appendix A, it can be seen that the values of $T_{\left(n\right)}{}^{L}$, $S_{\left(1\right)}{}^{NL}$ and $F_{\left(1\right)}$ are directly related to $u_{n}{}^{\left(1\right)}$ and $u_{n}$. When the fundamental frequency mode shape $u_{n}{}^{\left(1\right)}$ and the second harmonic mode shape $u_{n}$ are introduced into Equation (10), the amplitude parameter ${\bar{A}}_{n}$ corresponding to the mode combination can be calculated.

In existing articles about nonlinear ultrasonic waves, the relative nonlinear coefficient β is usually used to measure the change in material properties. The equation for β is:(11)$$\beta =\frac{A_{2}}{A_{1}{}^{2}},$$
where $A_{1}$ and $A_{2}$ represent the amplitudes of the fundamental frequency and the second harmonic signal. Since the amplitude of the fundamental frequency mode will not change during theoretical derivation, the value of β is mainly determined by the amplitude of the second harmonic mode. Therefore, there is a positive correlation between ${\bar{A}}_{n}$ and β. The variation law of ${\bar{A}}_{n}$ in theoretical derivation can reflect the change of β in physical experiments.

## 3. Response Law of Nonlinear Guided Waves to Temperature

According to the above analysis, the value of ${\bar{A}}_{n}$ is directly decided by the mode shapes of $u_{n}{}^{\left(1\right)}$ and $u_{n}$. When the mode combination of the fundamental frequency mode and the second harmonic mode is determined, the amplitude coefficient ${\bar{A}}_{n}$ is also confirmed. Compare ${\bar{A}}_{n}$ under different temperatures, the response law of the second harmonic modes with temperature can be theoretically derived.

### 3.1. Selection of Mode Combination

In this paper, the semi-analytical finite element (SAFE) method is adopted to deal with mode shapes, and the plate structure is taken as the research object. The reason for choosing the plate structure as the verification object is that there are few ultrasonic guided wave modes propagating in the plate, there are clear analytical solutions of ultrasonic guided waves in the plate, and it is convenient to carry out physical experiments.

Assuming there is an infinite-width steel plate model, the plate width along the y direction is infinite, and the plate thickness along the z direction is 15 mm. The propagation direction of guided waves is the x direction. One-dimensional three-node elements are used to discretize the cross-section of the model, and each node has three degrees of freedom. The material parameters of the plate are shown in Table 1.

Based on Hamilton’s theory, the general wave equation can be reduced to [27,28]:(12)$$\left[K_{1}+ikK_{2}+k^{2}K_{3}-\omega^{2}M\right]u=0,$$
where M is the mass matrix, $K_{1}$, $K_{2}$, and $K_{3}$ are the element stiffness matrices. Equation (12) can be solved as a linear generalized eigenvalue problem with a given wavenumber. The eigenvalue is $\omega^{2}$, and the eigenvector is mode shape u. After getting the relationship between mode shape, wavenumber, and angular frequency, the dispersion curves of phase velocity $C_{p}$ and group velocity $C_{g}$ can be drawn. The calculation formulas for phase velocity $C_{p}$ and group velocity $C_{g}$ are as follows:(13)$$C_{p}=\frac{\omega }{k},C_{g}=\frac{d\omega }{dk}.$$
The phase velocity and group velocity dispersion curves of ultrasonic guided wave in isotropic free steel plate are depicted by MATLAB R2021a software, as shown in Figure 1. Every data point in Figure 1 indicates one mode, and fd is the frequency-thickness product. From Figure 1, the phase velocity, group velocity, and mode shape of each ultrasonic guided wave mode can be acquired.

In order to investigate the influence of temperature on different cumulative second harmonics, it is necessary to first clarify the mode combinations that can generate cumulative second harmonics. In this paper, the ultrasonic excitation frequency is set as 2.5 MHz and the received frequency is 5 MHz. In terms of the principle in Equation (9), all the mode combinations that can produce cumulative second harmonics at 37.5 MHz-mm and 75 MHz-mm in Figure 1a are selected. Substitute the mode shapes of each mode combination into Equation (10) to calculate the value of ${\bar{A}}_{n}$, the relevant values of all mode combinations are shown in Table 2 [12].

It can be found from the data in Table 2 that, mode combinations 1–15 all satisfy the phase velocity matching condition, and the ${\bar{A}}_{n}$ of mode combinations 7–15 are not zero. So only mode combinations 7–15 can produce cumulative second harmonics. At the same propagation distance, the amplitudes of cumulative second harmonics modes produced by different mode combinations are not the same. Since the amplitude parameter ${\bar{A}}_{n}$ of mode combination 10 is the maximum, the cumulative second harmonic mode produced by mode combination 10 has the highest amplitude.

After determining the mode combinations, by substituting the related mode shapes under different temperature conditions into Equation (10), the variation of ${\bar{A}}_{n}$ with temperature reflects the response law of the cumulative second harmonic modal amplitude with temperature.

### 3.2. Theoretical Analysis Result

Presume the steel plate model is under normal situations and simply influenced by temperature. The temperature change range of the steel plate is –20 °C to 60 °C and the temperature rises by 5 °C each time. The normal atmospheric temperature is set at 20 °C. As the steel plate temperature rises from –20 °C to 60 °C, the material parameters E, λ and μ all change depending on the temperature state, and ν will not change during the entire process. When the plate model is at different temperature statuses, the current material parameters are listed in Table 3. Based on the data in Table 3, $K_{1}$, $K_{2}$, $K_{3}$ and M at a certain temperature can be calculated. Equation (12) is converted to the following expression when the temperature of the plate model is T:(14)$${\left[K_{1}+ikK_{2}+k^{2}K_{3}-\omega^{2}M\right]}_{T}u=0.$$
By solving Equation (14), the dispersion curves and mode shapes of ultrasonic guided waves at certain temperature states are acquired. The phase velocity dispersion curves at the temperature of T=20 °C and T=60 °C are shown in Figure 2a. The blue curve in Figure 2a represents the dispersion curve when the temperature is not applied, that is, the steel plate temperature is T=20 °C. The red curve in Figure 2a represents the dispersion curve when the steel plate temperature is T=60 °C. Magnify the dispersion curves in the circle in Figure 2a, as shown in Figure 2b. It can be seen from Figure 2b that when the temperature of the plate increases, the phase velocity of the ultrasonic guided waves mode decreases with the increase in temperature. On the contrary, when the temperature decreases, the phase velocity increases. After the application of temperature, the corresponding mode shapes of the ultrasonic guided waves undergo partial changes.

By obtaining different dispersion curves and mode shapes of the steel plate model under temperature, the change curves of amplitude parameters ${\bar{A}}_{n}$ of different mode combinations can be approximately described. Due to the proven ability of mode combinations 7–15 to generate cumulative second harmonics, mode combinations 7–15 are selected as the analysis objects in this section. The second-order curves are used to fit the change curves about ${\bar{A}}_{n}$ of mode combinations 7–15, and the theoretical analysis result is shown in Figure 3.

In Figure 3, the red star data points represent the amplitude parameters ${\bar{A}}_{n}$ of mode combinations, and the black curves show the change trends of ${\bar{A}}_{n}$. As shown in Figure 3, there are only two obvious tendencies in all the variation curves, monotonically increasing or monotonically decreasing. In the vary course of steel plate temperature rises from –20 °C to 60 °C, the amplitude parameters of mode combinations 8, 10, and 11 decrease gradually, while the amplitude parameters of mode combinations 7, 9, 12, 13, 14, and 15 increase gradually. The variation curve of the amplitude parameter indicates the changing trend of the corresponding cumulative second harmonic amplitude. Since there is only a temperature variable in the theoretical derivation process, the result in Figure 3 shows the response law of nonlinear ultrasonic guided waves under temperature action.

The theoretical results in Figure 3 present that, for different mode combinations, the cumulative second harmonic amplitudes produced by them exhibit different change trends in the same temperature change process, which are monotonically decreasing or increasing. In Figure 3, some data points are inconsistent with the overall vary tendency. The above phenomenon is caused by a fitting error when using the second-order functions to fit curves. The research purpose of this paper is to distinguish the variation trends of cumulative second harmonics with temperature, rather than obtaining an exact relationship equation. The presence of these data points will not affect the overall change trend prediction.

It is worth emphasizing that the mode combination selection method proposed in this paper is also applicable to other structures with arbitrary complex cross-sections and has a wide range of structural applicability. Through establishing the corresponding finite element model, the mode combinations that can generate cumulative second harmonics in the waveguide will be obtained, and further research about the influence of temperature or stress can be developed.

## 4. Simulation Experiments

In order to verify the correctness of the theoretical result, three-dimensional (3D) simulation experiments are conducted based on ABAQUS 2021 software. The finite element method (FEM) is a widely employed method in the field of structural imperfection detection [29], and it also exhibits obvious advantages in analyzing complex geometric structures [30] and providing reliable opinions on the nonlinear characteristics caused by defects. To further improve the reliability of numerical simulation results, 3D FEM has been applied to nonlinear ultrasonic research. Guan [31] dissected the contact acoustic nonlinearity in a cracked pipe model through a finite element analysis. Xu [32] established a 3D fatigue crack growth model based on contact acoustic nonlinearity to predict the continuous growth of the identified fatigue cracks along the length and depth. Lee [33] undertook the 3D simulation of nonlinear ultrasonic waves for fatigue damage detection using the precise fatigue crack trajectory. The 3D FEM simulation of nonlinear ultrasonic guided waves is also explored in this section.

By analyzing the data in Table 2, it can be found that cumulative second harmonic amplitudes excited by mode combinations 10 and 15 are the largest. More importantly, the change tendencies of the amplitude parameters of the two combinations are opposite with increasing temperature. The amplitude parameter of mode combination 10 increases monotonously, while the amplitude parameter of mode combination 15 decreases monotonously, which can form a good contrast. Therefore, mode combinations 10 and 15 are selected as the excitation and received second modes in verification tests. The mode shapes of mode combinations 10 and 15 are obtained by solving Equation (12), as shown in Figure 4.

In order to excite the selected second harmonics, the critical angular refraction method is used for ultrasonic excitation. The excitation and reception of the selected ultrasonic guided wave modes can be achieved by changing the incidence and acceptance angles. The incidence and acceptance angles θ of ultrasonic waves satisfy Snell’s theorem, which is the following:(15)$$\sin\theta /C_{\mathrm{glass}}=\sin90^{\circ }/C_{p},$$
where $C_{\mathrm{glass}}$ represents the propagation velocity of the ultrasonic guided wave in organic glass and $C_{\mathrm{glass}}=2740\;m/s$. The phase velocities $C_{p}$ of mode combinations 10 and 15 are $C_{p1}=3933\;m/s$ and $C_{p2}=3376\;m/s$. The incidence and acceptance angles of mode combinations 10 and 15 can be obtained by solving Equation (15), $\theta_{1}=44^{\circ }$ and $\theta_{2}=55^{\circ }$.

### 4.1. Model Establishment

The size of the established plate model is 100 mm × 5 mm × 15 mm, and the material parameters of the plate are consistent with the theoretical model. The plate model is discretized by hexahedral elements through ABAQUS software, and the mesh size should meet the following formula:(16)$$l\leq \frac{\lambda_{\min}}{10}=\frac{1}{10}\frac{C_{p}}{f_{\max}},$$
where l is the mesh size, $f_{\max}$ is the maximum excitation frequency, and $\lambda_{\min}$ is the corresponding minimum wavelength. Based on the data in Table 2, it can be calculated that l≤0.135 mm, so the mesh size of the model is determined to be 0.13 mm. To avoid interference from reflected signals, an absorption boundary with a width of 0.5 mm is set around the periphery of the plate, which is achieved by setting infinite elements at the model boundary. The model element type is set to C3D8R, and the absorption boundary element type is set to CIN3D8.

A 20-cycle sine wave pulse signal modulated by a Hanning window is used as the ultrasonic excitation signal, and the center frequency of the signal is 2.5 MHz, as shown in Figure 5. The initial distance between the excitation position and the receiving position is 30 mm. According to the MOSER principle, the integration time step is set to $2\times 10^{-8}\;s$ and the analysis step time is set as 0.00012 s.

### 4.2. Simulation Analysis Result

Maintain the incidence angle $\theta_{1}=44^{\circ }$ at the excitation node, and the received signal of mode combination 10 is shown in Figure 6a. Apply the Fast Fourier transform (FFT) on the received signal to abstract the fundamental frequency and the second harmonic signals. The natural logarithm of the spectrum analysis result is shown in Figure 6b, and there is a significant second harmonic at the frequency of 5 MHz. Gradually increase the propagation distance between the excitation and receiving positions, and calculate the corresponding relative nonlinear coefficient β. The change of the relative nonlinear coefficient β is shown in Figure 6c. Change the incidence angle $\theta_{2}=55^{\circ }$, and repeat the above simulation experiment. The simulation result of mode combination 15 is shown in Figure 7. The star data points in Figure 6c and Figure 7c represent the relative nonlinear coefficient. The variation curves in Figure 6c and Figure 7c show that the relative nonlinear coefficients all increase with increasing propagation distance, indicating the generation of cumulative second harmonics.

Apply temperature field influence to the model through ABAQUS software. The temperature field gradually increases from –20 °C to 60 °C with a change interval of 5 °C, and when the model is in different temperature states, modify the model material parameters according to the data in Table 3. Calculate the relative nonlinear coefficients of mode combinations 10 and 15 at different temperature conditions, and the simulation result is shown in Figure 8. The star data points in Figure 8 represent the relative nonlinear coefficient. It can be seen from Figure 8 that the relative nonlinear coefficient of mode combination 10 increases with increasing temperature, while the relative nonlinear coefficient of mode combination 15 decreases with increasing temperature. The above simulation result is consistent with the theoretical result in Figure 3d,i. The 3D FEM can provide clear simulated ultrasonic signal waveforms and calculate relative nonlinear coefficients, which can be directly contrasted with physical verification experiments.

## 5. Physical Demonstration Experiments

The relevant physical demonstration experiments are projected on a steel plate. The size of the steel plate is 2 m × 20 cm × 15 mm. The material parameters of the steel plate are the same as those in Table 1. The incidence and acceptance angles of ultrasonic excitation and receiving transducers can be adjusted. The center frequencies of the excitation and the receiving ultrasonic transducers are 2.5 MHz and 5 MHz. In the experiments, the excitation and reception of ultrasonic guided waves signal are realized by adopting RITEC RAM-5000 SNAP nonlinear high-energy ultrasonic testing system, the testing system is shown in Figure 9a. The longitudinal wave transducers are placed on the steel plate, as shown in Figure 9b. The excitation signal still adopts the 20-cycle sine wave pulse signal modulated by the Hanning window.

With the same excitation amplitude, adjust the inclination angle of transducers, and the selection of mode combination is changed. Set the incidence and the acceptance angles $\theta_{1}=44^{\circ }$, the received signal about mode combination 10 is shown in Figure 10a, and the natural logarithm of the spectrum analysis result is shown in Figure 10b. It should be noted that, owing to multi-modal characteristics, the generated second harmonics may not be unique, but only cumulative second harmonics play a dominant role. In the test to check whether the second harmonic signal generated by mode combination 10 is the cumulative second harmonic, the propagation distance gradually increases from the initial 95 mm to 155 mm. When the propagation distance increases by 10 mm, 200 samples are collected through the testing system, and the average value and standard deviation of the relative nonlinear coefficient β are calculated. The change curve of β with the increasing propagation distance is shown in Figure 10c. Change the angles of excitation and receiving transducers $\theta_{2}=55^{\circ }$, and repeat the above experiment. The experimental result of mode combination 15 is shown in Figure 11.

The experimental result in Figure 10c and Figure 11c shows that the detected relative nonlinear coefficients both rise with the increasing distance, which illustrates that mode combinations 10 and 15 all can produce cumulative second harmonics. The relative nonlinear coefficient measured by mode combination 10 is larger than that of mode combination 15, indicating that mode combination 10 can excite the cumulative second harmonic with higher amplitude. The above experimental result is consistent with theoretical and simulation conclusions, which proves the accuracy of the mode combination selection method.

After completing the above experimental operation, place the steel plate in a temperature control box. The temperature control box in this paper is originally used to heat rail, but in this section, it is applied to change the temperature of the steel plate. The air temperature in the temperature control box can be adjusted through the control panel. The temperature of the steel plate can reach a maximum of 60 °C and a minimum of –20 °C through the temperature control box. The temperature control box is shown in Figure 12a. The contact temperature sensor is used to monitor the temperature of the steel plate with a propagation distance of 145 mm. The block diagram of the physical experimental system is shown in Figure 12b.

Since the viscosity of common coupling gel will change greatly during the heating process, which will affect the transmission of the ultrasonic signal, AB mucilage is adopted as ultrasonic coupling gel in the experiments. The solidified AB mucilage has no viscosity, and the effects of temperature on it can be ignored. The average time for the steel plate temperature to increase by 1 °C is 30 min. When the temperature of the steel plate rises by 5 °C, 200 samples are collected through the testing system at high speed for less than 1 min, so the measured relative nonlinear coefficient in this time stage can indicate the ultrasonic nonlinear state at this temperature. Calculate the average value and standard deviation of the relative nonlinear coefficient β at different temperatures and obtain the change curve of β with temperature, as shown in Figure 13.

From the change curves in Figure 13, it can be observed that the relative nonlinear coefficients of mode combinations 10 and 15 have the opposite change trends during the same heating process. The relative nonlinear coefficient of mode combination 10 increases monotonically, while that of mode combination 15 decreases monotonically. The physical experimental result is consistent with the theoretical derivation and simulation analysis result, which proves the accuracy of the variation law.

The experimental result indicates that temperature has a significant impact on nonlinear ultrasonic guided waves propagating in metal waveguides, and even a temperature variation of 5 °C can cause a clear change of relative nonlinear coefficient. Consequently, when conducting research about nonlinear ultrasonic guided waves, it is necessary to obtain the influence of temperature on the selected cumulative second harmonic in advance and correct the test result accordingly.

## 6. Conclusions

Nonlinear ultrasonic guided waves have demonstrated excellent application potential in the current NDT research field, but the diversity and complexity of the modal response to temperature restrict the related applications. In this paper, plate structure is selected as the research model and the temperature effects on nonlinear ultrasonic guided waves are methodically and comprehensively analyzed. The main contributions are as follows:(1)An accurate and clear method for identifying the response law of different cumulative second harmonics to temperature is described in detail. Through calculating amplitude parameters based on the SAFE method, the mode combinations of the fundamental frequency mode and the second harmonic mode in the waveguide that can generate cumulative second harmonics can be confirmed, and the magnitudes of different cumulative second harmonic modes are quantified. The variation law of nonlinear guided waves to temperature is further determined by analyzing the amplitude parameters at different temperature states.(2)The theoretical derivation result indicates that temperature has significant effects on nonlinear ultrasonic guided waves. When the temperature of the steel plate increases monotonically, the variation trends of relative nonlinear coefficients about different cumulative second harmonic modes are not the same, showing monotonically increasing or decreasing. Therefore, in the research and experiments of nonlinear ultrasonic guided waves, especially when the temperature range of the tested object is large, such as the detection of thermal stress in rail, necessary temperature compensation or identification must be carried out on the detection result; otherwise, the incorrect result may be obtained.(3)The methods and conclusions proposed in this paper are also appropriate for the waveguide with complex cross-sections and have universal applicability. By combining the 3D FEM, the further study of nonlinear ultrasonic guided waves under the influence of other types of defects, such as micro-cracks and corrosion creep, can be developed. The main obstacle in practical application is the separation and extraction of specific second harmonic modes in complex cross-sectional structures, and more work is needed to achieve the above goals.

## Figures and Tables

**Figure 1 materials-16-03548-f001:**
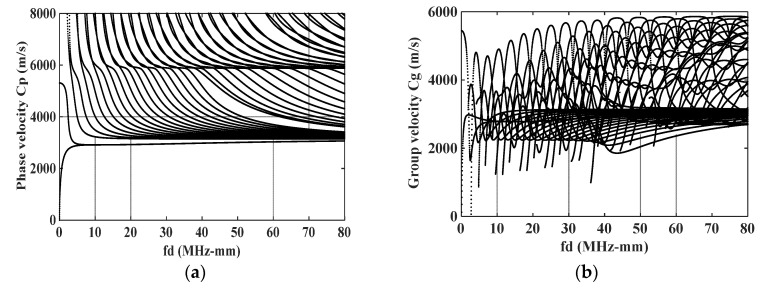
Phase velocity and group velocity dispersion curves: (**a**) Phase velocity dispersion curves; (**b**) Group velocity dispersion curves.

**Figure 2 materials-16-03548-f002:**
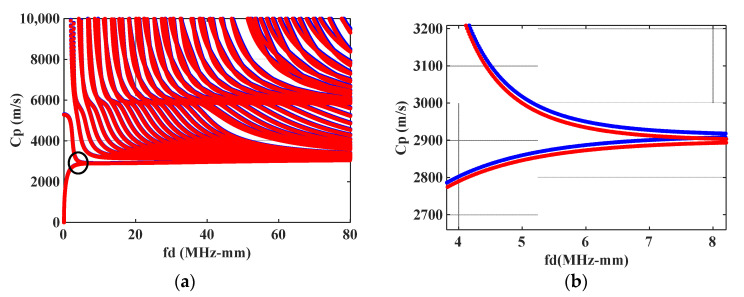
Phase velocity dispersion curves at different temperatures in steel plate: (**a**) Phase velocity dispersion curves; (**b**) Partial enlarged drawing.

**Figure 3 materials-16-03548-f003:**
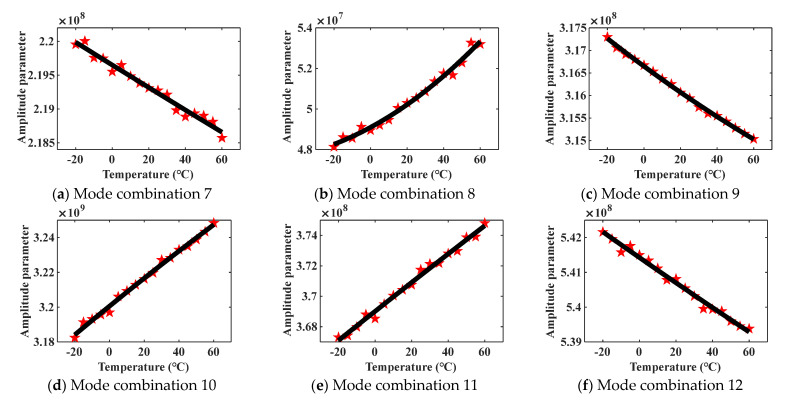
Amplitude parameters vs. temperature curves.

**Figure 4 materials-16-03548-f004:**
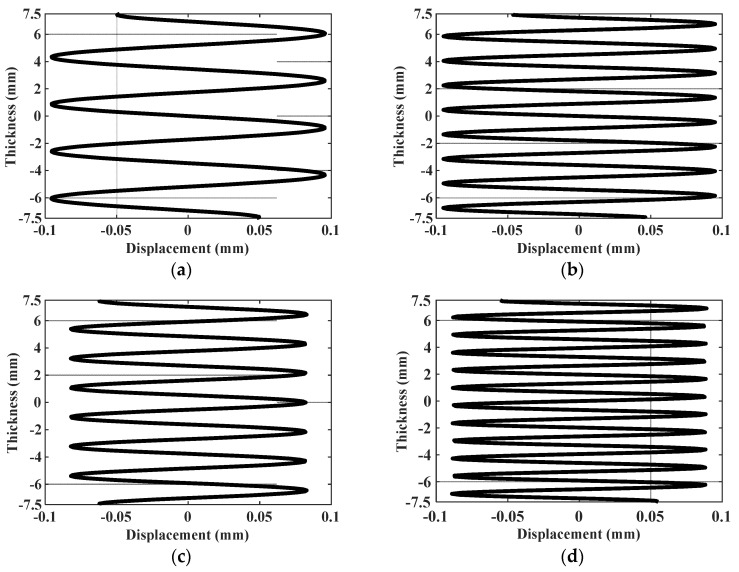
The mode shapes of ultrasonic guided waves in plate model: (**a**) The fundamental mode of mode combination 10; (**b**) The second harmonic mode of mode combination 10; (**c**) The fundamental mode of mode combination 15; (**d**) The second harmonic mode of mode combination 15.

**Figure 5 materials-16-03548-f005:**
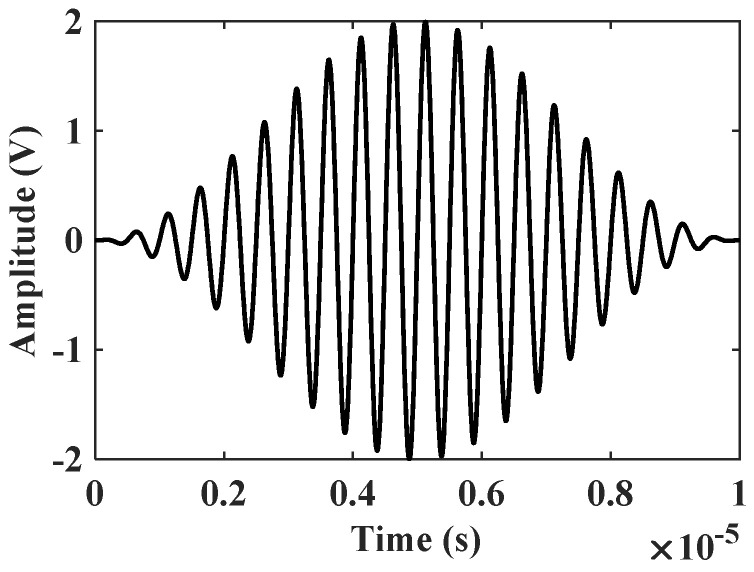
Excitation signal diagram.

**Figure 6 materials-16-03548-f006:**
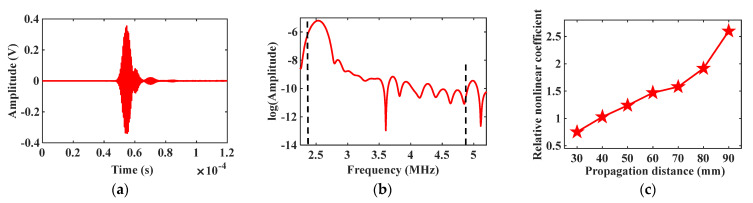
Simulation result of mode combination 10: (**a**) Received signal diagram; (**b**) Spectrum analysis result; (**c**) Relative nonlinear coefficient vs. propagation distance curve.

**Figure 7 materials-16-03548-f007:**
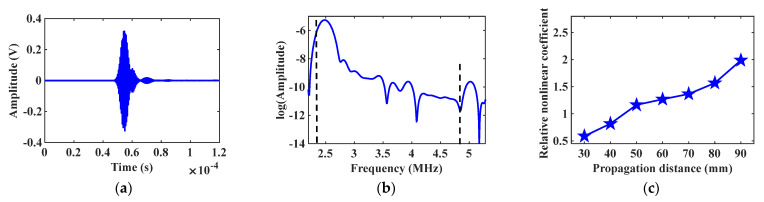
Simulation result of mode combination 15: (**a**) Received signal diagram; (**b**) Spectrum analysis result; (**c**) Relative nonlinear coefficient vs. propagation distance curve.

**Figure 8 materials-16-03548-f008:**
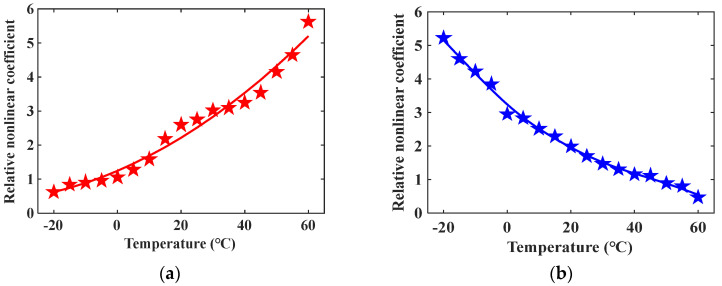
Simulation result of relative nonlinear coefficient vs. temperature: (**a**) Mode combination 10; (**b**) Mode combination 15.

**Figure 9 materials-16-03548-f009:**
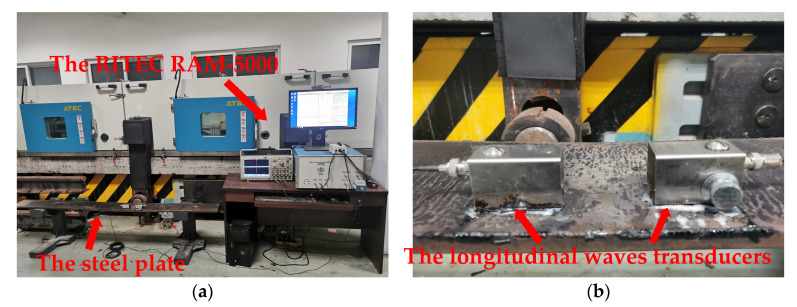
Testing system: (**a**) RITEC RAM-5000 SNAP; (**b**) Longitudinal wave transducers.

**Figure 10 materials-16-03548-f010:**
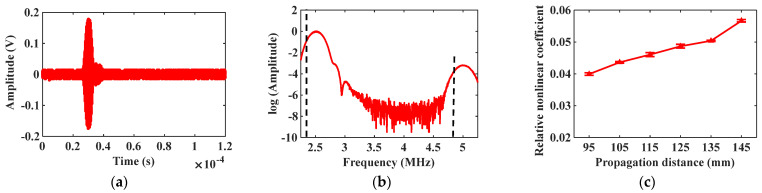
Experimental result of mode combination 10: (**a**) Received signal diagram; (**b**) Spectrum analysis result; (**c**) Relative nonlinear coefficient vs. propagation distance curve.

**Figure 11 materials-16-03548-f011:**
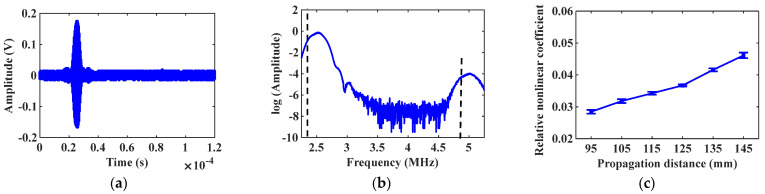
Experimental result of mode combination 15: (**a**) Received signal diagram; (**b**) Spectrum analysis result; (**c**) Relative nonlinear coefficient vs. propagation distance curve.

**Figure 12 materials-16-03548-f012:**
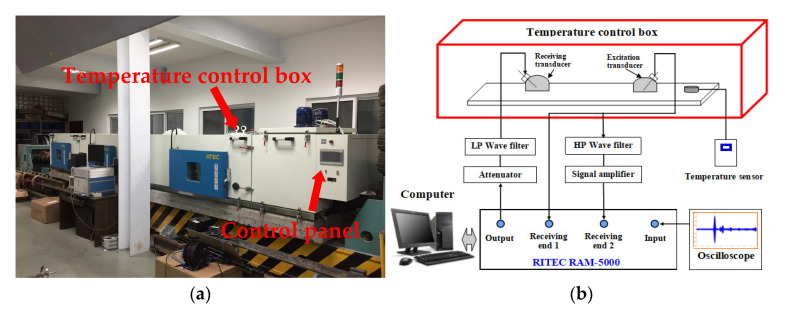
Physical verification experimental system: (**a**) Experimental temperature control box; (**b**) Block diagram of the physical experimental system.

**Figure 13 materials-16-03548-f013:**
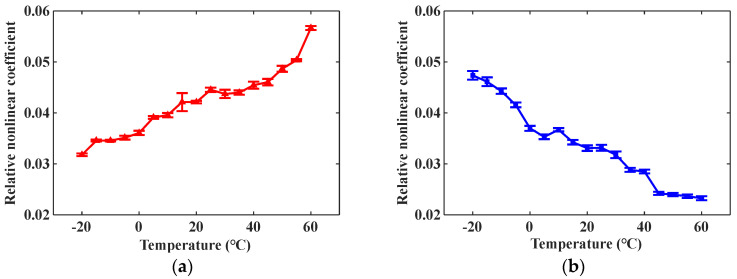
Experimental result of relative nonlinear coefficient vs. temperature: (**a**) Mode combination 10; (**b**) Mode combination 15.

**Table 1 materials-16-03548-t001:** The material parameters of the steel plate model.

Density	Elastic Modulus	Lame Coefficients		Poisson Ratio	Third-Order Elastic Coefficients		
$\rho (\mathrm{kg}/m^{3})$	*E* (GPa)	*λ* (GPa)	*μ* (GPa)	*ν*	*A* (GPa)	*B* (GPa)	*C* (GPa)
7932	200	115.38	76.93	0.3	–340	–647	–17

**Table 2 materials-16-03548-t002:** The relevant values of mode combinations.

Number	1		2		3		4		5	
fd (MHz-mm)	37.5	75	37.5	75	37.5	75	37.5	75	37.5	75
$C_{p}$ (m/s)	3144	3141	3144	3144	3154	3160	3171	3180	3195	3206
$C_{g}$ (m/s)	3013	2771	3013	3505	1372	2741	1294	2617	1361	2706
${\bar{A}}_{n}$($\times 10^{7}$)	0		0		0		0		0	
Number	6		7		8		9		10	
fd (MHz-mm)	37.5	75	37.5	75	37.5	75	37.5	75	37.5	75
$C_{p}$ (m/s)	5904	5908	3171	3169	3144	3147	3195	3192	3933	3920
$C_{g}$ (m/s)	2132	4917	1294	2744	3013	2765	1361	3142	1088	2492
${\bar{A}}_{n}$($\times 10^{7}$)	0		21.93		5.03		31.61		321.66	
Number	11		12		13		14		15	
fd (MHz-mm)	37.5	75	37.5	75	37.5	75	37.5	75	37.5	75
$C_{p}$(m/s)	3448	3458	3227	3223	3267	3263	3316	3315	3376	3382
$C_{g}$(m/s)	1271	2331	1343	2728	3013	2644	1303	2581	1275	2488
${\bar{A}}_{n}$($\times 10^{7}$)	37.08		54.08		82.13		113.38		133.26	

**Table 3 materials-16-03548-t003:** The material parameters at different temperature statuses.

T (°C)	E (GPa)	λ (GPa)	μ (GPa)	ν
60 °C	198	114.23	76.15	0.3
55 °C	198.25	114.38	76.25	0.3
50 °C	198.5	114.52	76.35	0.3
45 °C	198.75	114.66	76.44	0.3
40 °C	199	114.81	76.54	0.3
35 °C	199.25	114.95	76.63	0.3
30 °C	199.5	115.1	76.73	0.3
25 °C	199.75	115.24	76.83	0.3
20 °C	200	115.38	76.93	0.3
15 °C	200.25	115.53	77.02	0.3
10 °C	200.5	115.67	77.12	0.3
5 °C	200.75	115.82	77.21	0.3
0 °C	201	115.96	77.31	0.3
–5 °C	201.25	116.11	77.4	0.3
–10 °C	201.5	116.25	77.5	0.3
–15 °C	201.75	116.39	77.6	0.3
–20 °C	202	116.54	77.69	0.3

## Data Availability

Data are contained within the article.

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
