# Peer review of "Temperature Effects on Nonlinear Ultrasonic Guided Waves"

_materials, 2023, doi:10.3390/ma16093548_

Round 1
Reviewer 1 Report
The main observations are listed below. The acceptance of the manuscript would depend on the revision. The author needs to provide a point-by-point response or provide a rebuttal.
1) The abstract should be briefly written to describe the purpose of the research, the principal results, and major findings. Authors should revise it.
2) This paper should be edited grammatically. Please check the whole manuscript for typo and punctuation mistakes, such as Definite and Indefinite Articles (a, an, the). It is recommended that the wordings and grammar of English should be rechecked throughout the present manuscript.
3) Mathematical description of solution methodology should be provided briefly.
4) You should double check the mathematical formulations. Some mathematical symbols are not defined.
5) For general readers, authors are encouraged to discuss other kind of works on FEM such as: [(a) “Microstructural/geometric imperfection sensitivity on the vibration response of geometrically discontinuous bi-directional functionally graded plates (2D-FGPs) with partial supports by using FEM”, Steel and Composite Structures, 45(5), 621-640.; (b) “Static bending and buckling analysis of bi-directional functionally graded porous plates using an improved first-order shear deformation theory and FEM”, European Journal of Mechanics - A/Solids, 96, 104743.].
6) Result and discussion sections are poor. You should add more explanation to this section. Expand the discussion to highlight the relevance and interest of this work for its aimed scientific community.
7) Conclusion section is poor. Some applications of the model and future scope should be included.
I believe that the above changes will certainly add value to the already well-documented contribution by the authors. With these modifications, I think this already very good article can be improved somewhat and will be better. I then welcome it for publication after revision.
No comments.
Reviewer 2 Report
In this paper, the authors proposed an analysis approach to obtain the response law by introducing the amplitude parameter to quantify the amplitude change based on the semi-analytical finite element method. They used simulation analysis and experimental verification on a plate structure as an example. The result showed that the variation trends of different cumulative second harmonic modes with temperature are distinct, and the amplitudes are monotonically increasing or decreasing with the increase of temperature. The main findings were that, in the applications of nonlinear ultrasonic guided waves, it is necessary to predict the change trend of the cumulative second harmonic amplitude under the action of temperature, and compensate the result for the influence of temperature. In general, the work is interesting. The simulation and experiments were well-designed and the results were promising. However, following issues should be addressed:
1. The title: “The Diversity of Temperature Effects…” is very confusing. The word “Diversity” should be omitted.
2. Equation (1): Reference should be added.
3. Line 183: “why” should be omitted.
4. Line 192: Table 1 title should be moved to page 7 right before the content of the table.
5. Table 3: Why was the case with T>60 (oC) not investigated? In addition, the whole table should be arranged in one page.
6. Figure 3: The reason for choosing the mode combinations should be explained. In addition, the whole figure should also be arranged in one page or divided into different figures.
7. Figure 5: What is the “us” unit in the x-axis title?
8. Figure 6 a, b, c should be arranged in one page.
9. Figure 9b. The color of the embedded texts should be changed or the texts should be rearranged in the figure for better visualization.
10. Figure 12: Main components of the control box should be noted.
Minor editing of English language required.
Round 2
Reviewer 2 Report
The authors have addressed all issues suggested by the Reviewer. The manuscript is recommended for the publication in the journal.